# A Comparative Study of Supervised Machine Learning Algorithms for the Prediction of Long-Range Chromatin Interactions

**DOI:** 10.3390/genes11090985

**Published:** 2020-08-24

**Authors:** Thomas Vanhaeren, Federico Divina, Miguel García-Torres, Francisco Gómez-Vela, Wim Vanhoof, Pedro Manuel Martínez-García

**Affiliations:** 1Division of Computer Science, Universidad Pablo de Olavide, 41013 Sevilla, Spain; vtho@alu.upo.es (T.V.); fdivina@upo.es (F.D.); mgarciat@upo.es (M.G.-T.); fgomez@upo.es (F.G.-V.); 2Faculty of Computer Science, University of Namur, 5000 Namur, Belgium; wim.vanhoof@unamur.be; 3Centro Andaluz de Biología Molecular y Medicina Regenerativa (CABIMER), CSIC-Universidad de Sevilla-Universidad Pablo de Olavide, 41092 Sevilla, Spain; 4Facultad de Ciencias y Tecnología, Universidad Isabel I, 09003 Burgos, Spain

**Keywords:** machine-learning, chromatin interactions, prediction, genomics, genome architecture

## Abstract

The role of three-dimensional genome organization as a critical regulator of gene expression has become increasingly clear over the last decade. Most of our understanding of this association comes from the study of long range chromatin interaction maps provided by Chromatin Conformation Capture-based techniques, which have greatly improved in recent years. Since these procedures are experimentally laborious and expensive, in silico prediction has emerged as an alternative strategy to generate virtual maps in cell types and conditions for which experimental data of chromatin interactions is not available. Several methods have been based on predictive models trained on one-dimensional (1D) sequencing features, yielding promising results. However, different approaches vary both in the way they model chromatin interactions and in the machine learning-based strategy they rely on, making it challenging to carry out performance comparison of existing methods. In this study, we use publicly available 1D sequencing signals to model cohesin-mediated chromatin interactions in two human cell lines and evaluate the prediction performance of six popular machine learning algorithms: decision trees, random forests, gradient boosting, support vector machines, multi-layer perceptron and deep learning. Our approach accurately predicts long-range interactions and reveals that gradient boosting significantly outperforms the other five methods, yielding accuracies of about 95%. We show that chromatin features in close genomic proximity to the anchors cover most of the predictive information, as has been previously reported. Moreover, we demonstrate that gradient boosting models trained with different subsets of chromatin features, unlike the other methods tested, are able to produce accurate predictions. In this regard, and besides architectural proteins, transcription factors are shown to be highly informative. Our study provides a framework for the systematic prediction of long-range chromatin interactions, identifies gradient boosting as the best suited algorithm for this task and highlights cell-type specific binding of transcription factors at the anchors as important determinants of chromatin wiring mediated by cohesin.

## 1. Introduction

Mammalian genomes stretch for more than two meters and are formed by around three billion base pairs that are tightly packed within the nucleus, which has a width on the order of micrometers. Strikingly, this level of compaction is compatible with a proper accessibility to the cellular machinery required for essential metabolic processes such as replication or transcription. Over recent years, it has become clear that such seemingly counterintuitive events can be explained by the architectural organization of the genome, which forms 3D structures with several levels of complexity [1,2]. Beyond nucleosome-nucleosome interactions, chromatin loops represent the smallest scale of genome organization. Loops bring distal genomic loci into close physical proximity and typically range from one to several hundreds of kilobases [3,4]. On a larger scale, chromosomes are spatially segregated into structures called topologically associating domains (TADs). TADs are blocks of chromatin where all pairs of loci interact with each other more frequently than with neighboring regions [5,6]. At a higher level, the interactions of TADs with one another make up megabase-scale structures that extend to whole chromosomes and are known as nuclear compartment [7]. Since there is not yet a well characterized biological delineation between such orders of genome organization, in this study we will use the term ’loop’ to refer indistinctly to chromatin interactions at any level of this hierarchy. Accordingly, we will refer to the pair of distal loci that are brought together as ’loop anchors’.

Recent findings have revealed that 3D genome organization is more complex than anticipated [3]. Indeed, genome architecture can vary between cell types and is dynamic during cell differentiation and development [8]. Evidence also points to an essential role of the 3D genome in the control of gene expression by allowing communications between promoters and distal enhancers [9,10,11,12]. In addition, the insulator protein CCCTC-binding factor (CTCF) and the ring-shaped cohesin complex have been shown to highly co-localize both at the borders of TADs and at the anchors of intra-TADs chromatin interactions in mammalian cells, which likely indicates that these factors work together to shape chromatin architecture [3,5,6,7]. A plausible hypothesis of how TADs and loops are formed postulates that cohesin extrudes DNA loops until it encounters an obstacle such as convergently oriented loop anchor DNA sequences actively bound by CTCF [13,14]. This hypothesis places cohesin as the main player in the so-called loop extrusion model [13,14,15,16]. Interestingly, recent evidence has revealed that cohesin is moved to CTCF sites by transcription [17], suggesting that RNA polymerase II (Pol II) might be a driving force for this mechanism.

Most of the advances in our understanding of how high order genome organization links to essential cellular metabolic processes comes from the development of Chromatin Conformation Capture (3C)-based technologies [18], which has provided the scientific community with high resolution genome-wide chromatin interaction maps for several mammalian cell types. Despite technical improvements, experimental profiling of such maps not only remains difficult and expensive, but also requires a remarkably high sequencing depth for the achievement of high resolution [3,19]. Therefore, in silico predictions that take advantage of the wealth of publicly available sequencing data emerges as a rational strategy to generate virtual chromatin interaction maps in new cell types for which experimental maps are still lacking. To date, several studies have been devoted to predict chromatin loops based on one-dimensional (1D) genomic information with accurate results [20,21,22,23,24,25]. In such works, authors have modeled loops using different designs and machine learning approaches. Accordingly, they reached different conclusions regarding which chromatin features are most predictive and whether information within the loops (away from the anchors) contributes to the predictive power.

Here, we model cohesin-mediated chromatin loops using an integrative approach based on ENCODE 1D sequencing datasets to test the performance of six different machine learning algorithms: decision trees (DT), random forests (RF), gradient boosting (XGBoost), Support Vector Machines (SVMs), multi-layer perceptron (MLP) and deep learning Artificial Neural Network (DL-ANN). We find that XGBoost achieves the best performance, with precision scores of around 0.95. We also show that, although architectural features at the anchors display the greatest predictive power, transcription-associated features accurately predict chromatin loops. Among these features, transcription factors are the most informative and their contribution resides mainly at the anchors and varies across human cell lines.

## 2. Materials and Methods

### 2.1. Processing Publicly Available Data

The experimental data used in this study is summarized in Appendix A. With the exception of Hi-C, which was used for visualization purposes, the rest of the datasets were batch-downloaded from ENCODE [26]. BAM files were used for ChIP-seq, RNA-seq and DNase-seq, while tsv tables labeled as ‘long range chromatin interactions’ were used for ChIA-PET experiments targeting RAD21. Figure 1 and Appendix A show genome browser views displaying a selection of these datasets. All the analyses were performed in K562 and GM12878 cell lines using human assembly hg19.

### 2.2. Identification of RAD21-Associated Loops

RAD21 loops in K562 were extracted from ENCODE dataset ENCFF002ENO. For the identification of RAD21 loops in GM12878, two replicates were considered (datasets ENCFF002EMO and ENCFF002EMQ) and only overlapping loops were retained. We defined overlapping loops as those that share both anchors, allowing a maximum gap of 2 kb. Following this approach, we identified 3290 and 5486 loops in K562 and GM12878, respectively. For machine learning classification, we generated the same number of negative loops for each cell line. In order to build robust models that were able to accurately separate loops from random genomic regions, we generated an initial set of negative loops by randomly combining pairs of RAD21 ChIP-seq peaks (see Figure 2a). Then, loops that overlapped experimental ones were filtered out. Since genomic distance highly influences chromatin interactions, the resulting set of negative loops were passed through a regression model trained to capture the distribution of genomic distances between anchors of experimental loops. Finally, 3290 and 5486 loops were randomly selected from the K562 and GM12878 negative sets, respectively. In this way, the number of positive and negative loops in the final datasets were equal and with similar genomic distance distributions. Since we used pairs of RAD21 peaks to generate the initial set of background loops, we examined RAD21 (and CTCF) ChIP-seq reads at the anchors of our positive and negative loops. We observed a clear read enrichment of both signals (Appendix A), indicating that final loop sets are adequate for subsequent training.

### 2.3. Machine Learning Data Matrix

To model loops using chromatin features, we quantified 23 sequencing datasets within and adjacent to each loop using a modified version of the approach described by Handoko et al. [27]. Given a loop with length L, we extended L base pairs to its left and right and the extended region (with length 3L) was splitted into 1500 bins (Figure 2b, top). Then, we scored the sequencing experiments (Appendix A) within each bin (Figure 2b, bottom). The scoring was performed by counting reads that aligned to each bin and normalizing by bin genomic length and sequencing library size. For each cell line, we obtained a final data matrix with rows representing loops and columns representing quantification of chromatin features at each bin.

The 23 chromatin feature datasets (Appendix A) included marks associated with chromatin accessibility (DNAse-seq), expression (RNA-seq), RNA Pol2 binding (POLR2A, POLR2AphosphoS5), active promoters (H3k4me2, H3k4me3, H3k9ac), enhancers (H3k4me1), active gene bodies and elongation (H3k36me3, H4k20me1), transcriptional repression (H3k27me3, H3k9me3), architectural components (CTCF, RAD21) and transcription factors (ATF3, CEBPB, JUND, MAX, REST, SIN3A, SP1, SRF, YY1). The motivation of selecting these datasets was to keep a wide set of chromatin features representing different molecular events. Only datasets from ENCODE common to both cell lines were selected.

### 2.4. Supervised Learning

In this work, we used six Machine Learning strategies in order to induce a classification model for the data. Each of the model is applied to a training set and then the prediction models are evaluated on a test set. We briefly describe each method in the following subsections.

#### 2.4.1. Decision Trees (DT)

We have used the algorithm implementation provided by [28,29]. A decision tree (DT) algorithm iteratively builds a classification tree by adding a node to the tree. Features are used within internal nodes in order to classify examples.

A DT algorithm induces a tree by splitting the original dataset into smaller sets based on a test applied to the features. This process is repeated recursively on each smaller set and is complete when the small set present in one node presents the same value as the target label or when no more gain in predictive power is obtained by splitting further. Such a process is known as recursive partitioning.

The features selected depend on a measure used in order to assess the importance of a feature. The Gini Impurity measure was used, which is defined as:(1)$$1-\sum_{i=1}^{n}p^{2}\left(c_{i}\right)$$
where $p(c_{i})$ s the probability of class $c_{i}$ in a node. We have also run several experiments on the datasets used in the paper with the Entropy, and the results were basically the same with the two measures.

In this paper, we used the implementation provided by the library Scikit Learn (Sklearn) library [28]. The maximum depth parameter has been set to 10. The criterion was set to Entropy, while Gini is in theory intended for continuous attributes, Entropy is intended for attributes occurring in classes but since Entropy uses logarithmic functions slower to compute. For the training of the K562 DT, both criteria were tested and compared.

#### 2.4.2. Random Forests (RF)

This strategy [30] belongs to the family of ensemble learning algorithms, based on a divide and conquer approach that improves performance. The principle behind ensemble methods is that a group of weak models are put together to form a stronger model. This is due to the fact that ensemble strategies reduce variance, improving the prediction power. RF algorithms induce a set of trees, which are then used in order to produce the final output, using a voting scheme. The trees are built one at a time. Each tree is obtained using a randomly selected training subset and a randomly selected subset of features. It follows that the trees depend on the values of an independently sampled input data set, using the same distribution for all trees.

In this case we also used the implementation provided in [29]. We have set the number of estimators to 250, even if we noticed that the results show little variation with estimators between 50 and 250. The maximum depth of the trees was set to 8, and the criteria used was the Entropy.

#### 2.4.3. XGBoost

This algorithm, as well as other gradient boosting algorithms such as the well known Gradient Boosting Method (GBM), sequentially assembles weak models to produce a predictive model [31]. During this sequential procedure, a gradient descend procedure is applied. This procedure is repeated until a given number of trees has been induced, or when no improvement is registered. An important aspect of XGBoost is how the algorithm controls over-fitting, which is a a known issue in gradient boosting algorithms. XGBoost adopts a more regularized model formalization, which allows the algorithm to obtain better results than GBM. XGBoost is a method that has recently received much attention in the data science community, as it has been successfully applied in different domains. This popularity is mostly due to the scalability of the method. In fact, XGBoost can run up to ten times faster than other popular approaches on a single machine. Again, we used the Scikit-learn library for this algorithm. In this case the number of estimators was set to 100, the learning rate to 0.1 and the maximum depth to 3.

#### 2.4.4. Deep Learning

Deep learning [32] is a subset of machine learning that incorporates computational models and algorithms that mimic the architecture of biological neural networks in the brain, such as Artificial Neural Networks (ANNs). In a rough analogy with biological learning system, ANNs consist of densely interconnected units, called neurons [33]. Each neuron receives several real valued input, e.g., from other neurons of the network, and produces a single real valued output. The output depends on an activation function used in each unit, which introduces non-linearity to the output. The activation function is used only if the input received by a unit is higher than a given activation threshold. If this is not the case, then no output is produced. Normally, an ANN consists of different layers of neurons. The term “Deep” refers to the number of such layers and complexity of an ANN. There are three types of layers: the input layer, the output layer, and the hidden layer (which extracts the patterns within the data). Therefore, as the data moves from one hidden layer to another, the features are recombined and recomposed into complex features. Because of this, deep learning works especially well with unstructured data, but requires a huge volume of training data. Deep learning has proven to be successful in different applications fields such as acoustics, imaging or natural language processing.

In this paper, we used the Multi-layer Perceptron (MLP) classifier provided by Sklearn library [28]. In particular, for the experiments performed on the K562 cellular line, we used an Alpha (regularization term) of 0.001, with an adaptive learning rate and a network consisting of 40 layers with 10 neurons on the first layer. Alpha forces the parameters to have small values for robustness. The optimal hyper-parameters used to train the MLP classifier for the GM12878 cell dataset were the same as for the K562 cell, except for the Hidden Layer Sizes which was set to 100 layers, with 20 neurons on the first layer. These parameters were experimentally determined.

We have also used the implementation offered by Keras library [34]. For K562 loops, we first tested a neural network made of two hidden layers: Layer 1, 120 neurons, relu activation; Layer 2, 10 neurons, relu activation; Layer 3 (output), 1 neuron, sigmoid activation. The resulting accuracy was 0.99 for training and 0.77 for testing, hinting at a possible overfit. We have also tested the same network architecture with a reduced dataset (removing features associated with RAD21 and CTCF), and obtained 0.99 for testing and 0.69 for testing. Since it is probable that the model has overfitted, we have decided to try out a separate architecture on the K562 cell line, that also included Dropout layers; specifically, the architecture looked like this: Layer 1, 120 neurons, relu activation; Layer D1, Dropout 0.2; Layer 2, 30 neurons, relu activation; Layer D2, Dropout 0.2; Layer 3, 10 neurons, relu activation; Layer 4 (output), 1 neuron, sigmoid activation. The results slightly improved, obtaining for the full dataset on training 0.99 and testing 0.79 (0.02 improvement). The results for the reduced dataset stayed the same.

For GM12878 loops, we obtained an accuracy of 0.99 on the full dataset for training and 0.81 for testing. When we tried the reduced dataset, the models yielded an accuracy of 0.99 for training and 0.66 for testing. Given the low difference between the accuracies of an architecture that includes Dropout layers, we decided not to train GM12878 on a dropout-enabled neural network. Then again, the proposed architecture probably overfits, but the accuracies obtained stay in line with the those obtained using other algorithms presented in this paper.

#### 2.4.5. Support Vector Machines (SVM)

This supervised technique tries to find an hyperplane in a *n*-dimensional space (being *n* the number of features), capable of separating the training examples [35]. Several such hyperplanes may exist, and SVM aims at finding the one that maximazes the distance between data points of the classes. The hyperplane is basically the classification model, with data falling on one side of the hyperplance being assigned a class, while data falling on the other side are assigned the other class. Data are not always linearly separable, and in these cases, a kernel function is used in order to transform the original data points with the aim of mapping the original data points into a higher dimensional space where we can find a hyperplane that can separate the samples.

In this study we used the Radial Basis Function (RBF) kernel. The RBF kernel is an often used kernel for classification tasks [36]. In particular, the C-value for the RBF function was set to 100, after having performed several preliminary experiments.

To evaluate the methods, we used four popular measures [37]:(2)$$Accuracy=\frac{TP+TN}{TP+TN+FP+FN}$$
(3)$$Precision=\frac{TP}{TP+FP}$$
(4)$$Recall=\frac{TP}{TP+FN}$$
(5)$$F1\text{-}Score=\frac{2\times Precision\times Recall}{Precision+Recall}$$

In the above equation we used the following terminology:True Positive (TP): a true positive represents the case when the actual class matches the predicted class;True Negative (TN): a true negative is similar to a true positive, the only difference being that the actual and predicted classes are part of the negative examples;False Positive (FP): a false positive is the case when the actual class negative, but the example is classified as positive;False Negative (FN): similar to false positive, but it this case this refers to the case where a positive example is classified as negative.

## 3. Results and Discussion

### 3.1. Association of Genomic and Epigenomic Features with Chromatin Loops

We started by assessing the genomic and epigenomic landscape of chromatin loops. We used published ChIA-PET datasets targeting the cohesin complex component RAD21 [38] from two human cancer cell lines (Appendix A), K562 and GM12878, identifying 3,290 and 5,486 chromatin loops, respectively. 1D sequencing datasets from ENCODE (ENCODE Project Consortium, 2012) were also collected in order to represent the chromatin features associated with loops and their genomic neighbourhoods (Figure 1 and Appendix A). As expected, we observed a large colocalization of CTCF and RAD21 architectural proteins with loop anchors, in agreement with previously reported data [5,39]. On the other hand, whereas the repressive mark H3K27me3 and the gene body mark H3K36me3 were found to be dispersed along loops and across loop anchors, other regulatory marks were enriched both at loop anchors and inside the loops. Such is the case of Pol2, open chromatin measured by DNase-seq and several transcription factors. The same stands for H3K4me1 and H3K4me3, two well-known markers for enhancers and promoters. These observations suggest that regulatory features measured by high-throughput sequencing provide a valuable source of information for the prediction of chromatin loops.

### 3.2. An Integrative Approach to Predict Chromatin Loops

To integrate sequencing features data into a predictive model of chromatin loops, we applied the computational framework of Figure 2. We first generated positive and negative sets of RAD21-associated chromatin loops. We used the experimental loops from the previous section as the positive set and the negative set was generated by combining pairs of RAD21 ChIP-seq peaks from ENCODE that do not overlap experimental loop anchors (see Material and Methods). Given the relative distribution of regulatory marks with respect to chromatin loops (Figure 1), we argued that these marks may affect loops not only at the anchors but also at the region between them. Therefore, to comprehensively measure the occupancy of chromatin features within and adjacent to each loop, we based our strategy on the approach of [27]. Given a loop with length L, we extended L base pairs to its left and right and the extended region (with length 3L) was partitioned into 1500 bins. For each bin, 23 high-throughput sequencing experiments (Appendix A) were scored, resulting in a data matrix with rows representing loops and columns representing the scored experiments at each bin. Finally, we trained and tested classifiers in both cell lines using six machine learning algorithms. This design based on multi bins allowed us to measure the position-specific ability of sequencing data to predict chromatin loops with an unprecedented resolution. As a result, we ended up with model matrices of 34,500 columns and either 3290 (K562) or 5486 (GM12878) rows (Figure 2).

### 3.3. Model Performance

The final feature matrices were divided into training (80%) and test (20%) and six classification algorithms were applied: decision trees, random forests, XGBoost, SVM, MLP and DL-ANN. To evaluate the performance of classification, trained models were applied to the test sets and several metrics were calculated including accuracy (Acc), precision, recall and F1-score. Accurate predictions were obtained by the six algorithms, with almost no differences on the performance metrics and with similar values in both cell lines (Table 1 and Table 2). The smallest accuracies were obtained by DL-ANN, with values of 0.81 and 0.79 for GM12878 and K562, respectively. On the other hand, XGBoost significantly outperformed the rest of the methods, achieving accuracies of 0.95 and 0.96 for GM12878 and K562, respectively. As a matter of fact, the other four classification algorithms achieved very similar performance for GM12878 cell line, with Acc ranging from 0.81 to 0.83, while for K562 they ranged from 0.82 to 0.87. These results demonstrate that chromatin loops can be accurately predicted using our integrative approach and indicate that gradient boosting by XGBoost provides the best performance for this task.

### 3.4. Loop Anchors Are the Most Informative Regions

Next, we explored the most informative features for predicting chromatin loops. The design of our approach based on Handoko et al. [27] represents each chromatin feature as a 1500-bin array (Figure 2), which allowed us to comprehensively evaluate the contribution of a given feature according to its relative position within and at both sides of loops. Among the six methods we compared, DT, RF and XGBoost assign an importance measure during the training process, providing information on which repertoire of features are the most relevant in the classification. For this reason, and given that these algorithms showed better overall performance than SVM and MLP (Table 1 and Table 2), the latter were excluded from this analysis. As expected, binding of architectural components (CTCF, RAD21) showed the highest predictive power in both cell lines and regardless the learning algorithm (Figure 3 and Appendix A). We observed that the relative positions of the most informative features are highly biased towards genomic bins close to loop anchors. This is true for the top 10 important bins of both cell lines and for the three tested algorithms, suggesting that chromatin information between anchors in only modestly predictive.

Although the three methods overall agreed on the top important features, while for DT the predictive power is concentrated in ∼10 bins, this is not the case of RF and XGBoost, where it seemed to be more widely distributed (Appendix A). Among the top 10 important features of each algorithm in GM12878, bins associated with CTCF and RAD21 binding around loop anchors were largely predominant (Figure 3, bottom panel). The only exception was a bin associated with the transcription factor ATF3 at the right anchors, which was identified as the 8th most important feature by RF. On the other hand, in K562 REST was found together with CTCF and RAD21 among the top 10 important features by the three methods (Figure 3, top panel). Bins associated with this transcriptional repressor at loop anchors, although less informative, are as frequent as those associated with CTCF and RAD21.

The contribution of features that do not belong to the top 10 important ones greatly varies from one method to another. For example, DT and XGBoost reported several bins associated with histone marks among the top 30 important features in both cell lines, while this was never the case for RF (Appendix A). On the other hand, RF seemed to assign some predictive importance to DNase-seq and, in a lesser extent, to YY1, whereas these features were rarely found among the top 30 important features when using DT and XGBoost. These observations are likely to indicate that the contribution of such features is negligable compared to the top important ones.

Our design also allowed us to represent the contribution of each chromatin feature according to its relative position within and at both sides of loops (Figure 4, Figure 5 and Appendix A). In agreement with the importance analysis, most of the predictive information for the majority of the evaluated chromatin datasets resides at the anchors and their close genomic vicinity. Besides the architectural components CTCF and RAD21, this is particularly prominent for DNase I hypersensitivity and transcription factors (Figure 4 and Figure 5). Altogether, these results suggest that, in addition to architectural components, transcriptional features at the anchors contribute to chromatin wiring in RAD21-mediated loops.

### 3.5. Removing Architectural Features Has a Modest Effect on XGboost Performance for K562

Given the importance of transcription associated factors in the final predictions, we next investigated whether these features alone can be used in order to predict chromatin loops. To this aim, we removed CTCF and RAD21 from the matrices and evaluated models trained with the rest of the datasets. We observed that the six algorithms decreased their performance, with significant differences in the acquired predictions and GM12878 showing more pronounced drops (Table 3 and Table 4). The highest decrease was reported by DT in GM12878 cell line, which yielded an accuracy of 0.68, while when the whole set of features was used, the accuracy achieved was of 0.83 (Table 2). SVM, MLP and DL-ANN performance in both cell lines also seemed to be drastically affected by the removal of architectural factors. While these methods exhibited accuracies of 0.79–0.83 in the previous analysis, these values decreased to 0.66–0.73 after removing CTCF and RAD21 ChIP-seq data. Although its performance sensibly decreased in GM12878, XGboost was found again to achieve the best predictions. Strikingly, the performance of this algorithm was almost not affected by the removal of architectural components in K562 cell line, yielding an accuracy of 0.95 (Table 3). This result agrees with our previous analysis of relative importance, in which the binding of REST transcription factor at loop anchors was found to be among the most predictive features. We can then conclude that, at least in K562, transcription information alone is enough for the prediction of RAD21-mediated chromatin interactions.

Exploration of the most informative features in the new DT, RF and XGBoost models reveals that most of the predictive power resides in transcription factor binding and, to a lesser extend, DNase I hypersensitivity and Pol II binding (Figure 6 and Appendix A). Again, we observed that the relative positions of these features were biased towards genomic bins in close proximity to loop anchors. Although different prediction accuracies were achieved by the three algorithms, overall the sets of top important features they provided were similar. Since XGBoost yielded the best performance, we focused on the importances reported by this algorithm. We observed that different transcription factors govern the contribution of the two cell lines. While bins associated with REST and MAX were predominant among the top 10 features in K562 (Figure 6, top and right), this was not the case in GM12878, for which ATF3 and CEBPB (together with DNAse-seq and Pol II) were found to contribute most to the predictions (Figure 6, bottom and right). Given the role that chromatin interactions play in gene regulation, these results agree with distinct gene regulatory programs being maintained through cell-type specific binding of transcription factors [40,41], and highlights transcriptional features as important determinants of chromatin wiring mediated by RAD21.

### 3.6. XGboost Models Achieve Accurate Predictions When Trained with Subsets of Chromatin Features

Since the removal of architectural components yielded different outcomes depending both on the algorithm type and the cell line, we next explored to what extent specific subsets of chromatin features provide different prediction abilities. To that aim, we trained and tested DT, RF and XGBoost models based on the following categories: *Architectural* (only bins associated with CTCF and RAD21), *TF* (bins associated with transcription factors ATF3, CEBPB, JUND, MAX, REST, SIN3A, SP1, SRF and YY1), *Architectural-anchors* (CTCF and RAD21 around loop anchors) and *TF-anchors* (TF around loop anchors). To only account for information around anchors, we restricted the model matrices to 100 bins centered at left and right anchors, respectively, obtaining 200 bins (out of the total 1500 bins; Figure 2b) for each chromatin feature belonging to the defined categories. As a matter of fact, we also included in the analysis the two categories explored in previous sections, which we named *All* (all bins, as in Section 3.3 and Section 3.4) and *Transcription* (TF + DNase-seq + RNA-seq + histone marks, as in Section 3.5).

For all the evaluated subsets, XGBoost yielded the best performance in both cell lines (Figure 7 and Table 1, Table 2, Table 3 and Table 4 and Appendix A), in agreement with our previous observations. We also observed that models trained with subsets restricted to bins around anchors achieved performance at least as accurate as those obtained using also bins within the loops, which confirms that genomic information away from the anchors poorly contributes to chromatin wiring prediction. Overall, different accuracies were observed for the grouped categories, with GM12878 models being more sensible to subset selection. In this sense, DT and XGBoost showed a similar behaviour (Figure 7, left and right panels and Table 2, Table 4, Appendix A). For both algorithms, performance of K562 models were found to be somehow stable across categories, achieving accuracies of 0.83–0.87 (DT) and 0.94–0.96 (XGBoost) (Figure 7, left and right panels and Table 1, Table 3, Appendix A). On the other hand, GM12878 models achieved greatly variable predictions for different subsets of features, with *TF* associated categories yielding the worst performance (0.660.68 and ∼0.79 for DT and XGBoost, respectively). Conversely, *Architectural* categories were found among the most predictive ones, with accuracies of ∼0.85 (DT) and ∼0.93 (XGBoost) (Figure 7, left and right panels and Table 2, Table 4, Appendix A).

Unlike DT and XGBoost, RF performance seemed to be more consistent across cell lines and categories, with K562 achieving slightly better predictions (Figure 7, center panel and Table 1, Table 2, Table 3 and Table 4 and Appendix A). *Architectural* associated categories yielded the best performance, with accuracies of 0.91–0.93 and 0.89–0.91 in K562 and GM12878, respectively. It is worth noting that *Architectural-anchors* not only outperformed *Architectural* category, but also models trained with all the datasets (*All*) in both cell lines. This is also true for DT (Figure 7, left panel), and highlights the importance of information around anchors for the prediction of long-range chromatin interactions.

As we mentioned before, XGBoost yielded the best predictions (Figure 7, right panel) in both cell lines. In the case of K562, high accuracies were achieved independently of the selected features, confirming our observation that, at least in this cell line, not only architectural but also transcription factors are important determinants of RAD21-associated chromatin wiring.

### 3.7. Prediction across Cell Types

We next asked to what extend our DT, RF and XGBoost models could be generalized from one cell line to another. Models trained on GM12878 and applied to K562 yielded overall good performance when using the whole set of chromatin features (Figure 8, right panel). Again, XGBoost outperformed the other two algorithms, obtaining an accuracy of 0.9. While DT showed a moderate accuracy of 0.75, RF seemed to be the worst suited method for cross cell line predictions, yielding an accuracy of 0.63. When evaluating the subsets of features described in the previous section, different predictive performance was observed for the grouped subsets. Since these differences were consistent across the three methods and XGBoost always obtained the best results, we focused our analysis on the accuracies provided by this algorithm (Figure 8, right panel, grey bar). While models trained with architectural factors achieved satisfactory accuracies (0.91–0.92), those trained with TF or transcription-associated features showed only modest prediction abilities (accuracies of 0.65–0.67). Given the overall high predictive power that transcriptional features showed when trained and applied on the same cell line (accuracies of 0.79–0.95), these observations highlight that, unlike architectural factors, transcriptional features are highly cell line specific in the context of chromatin wiring prediction.

Although a similar pattern of performance was observed for the different subsets of features when we trained models on K562 and applied to GM12878, this time prediction accuracies dramatically decreased to 0.53–0.64 (Figure 8, left panel). Since the remarkable performance obtained for models trained and tested on K562 were evaluated on data matrices that were not used for training, we discard a potential overfiting within this cell line. However, the significant differences observed in cross cell line applications of GM12878 and K562 models suggest that the latter might be overfitting the cell line specific chromatin feature associations with RAD21 chromatin wiring. Therefore, although the results derived from our K562 models are consistent with those obtained using GM12878 and with previous findings [22,23,24,25], we conclude that these models are not adequate for cross cell line predictions.

## 4. Conclusions

In light of our results, we propose XGBoost as the best suited algorithm for the prediction of long-range chromatin interactions. According to our data, XGBoost can be used to generate genome-wide maps of chromatin interactions, and information on a few chromatin features at the anchors may be enough to yield accurate predictions. For cross cell line application of the predictive models, architectural factors alone appear to be sufficient, while transcriptional features do not seem to have enough predictive ability, suggesting that they are highly cell line specific in the context of chromatin wiring. However, examination of other cell lines and tissues is needed to confirm these observations. Similarly, constructing generalizing models trained with datasets from several cell lines would overcome potential overfitting. In addition, analysis of interactions mediated by other proteins will help to clarify whether the observed performance for RAD21 mediated wiring in K562 can be generalized. Finally, prediction of de novo long-range chromatin maps genome-wide and subsequent comparisons with experimental data can be helpful to more comprehensively assess the predictive power of our strategy, as well as to exploit its full predictive potential.

## Figures and Tables

**Figure 1 genes-11-00985-f001:**
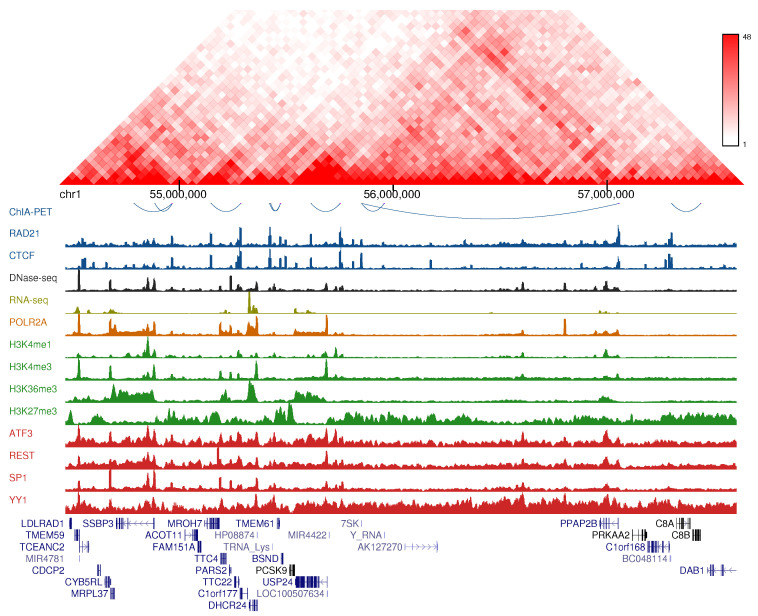
Chromatin features associated with RAD21 loops in GM12878 cell line. Hi-C interaction frequencies are shown in the top panel and ChIA-PET interactions are represented as blue arcs in the second panel. Then, a genome browser view for relevant chromatin features is displayed, including architectural factors (blue), DNase-seq (black), RNA-seq (yellow), RNA Pol2 (orange), histone marks (green) and transcription factors (red).

**Figure 2 genes-11-00985-f002:**
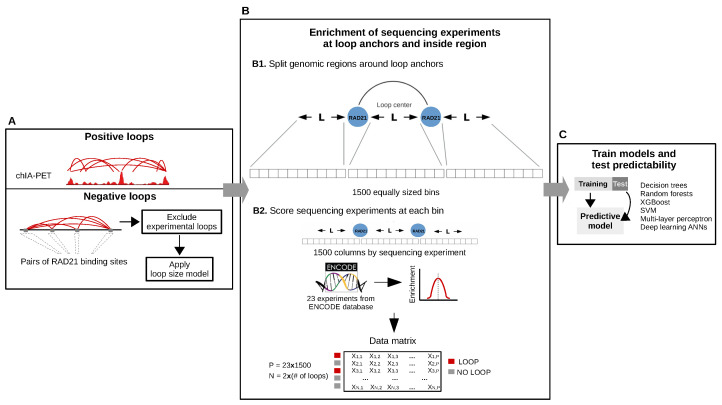
Illustration of the integrative machine learning schema for the prediction of chromatin loops. (**A**) Positive and negative RAD21-associated loops were first identified (see Methods). (**B**) Then, given a loop with length L, we extended L base pairs to its left and right and the extended region (with length 3L) was partitioned into 1500 bins. For each bin, 23 high-throughput sequencing experiments were scored, resulting in a data matrix with rows representing loops and columns representing the scored features. (**C**) Finally, we trained and tested classifiers using six machine learning algorithms. XGBoost: Gradient boosting; SVM: Support Vector Machines; ANN: Artificial Neural Networks.

**Figure 3 genes-11-00985-f003:**
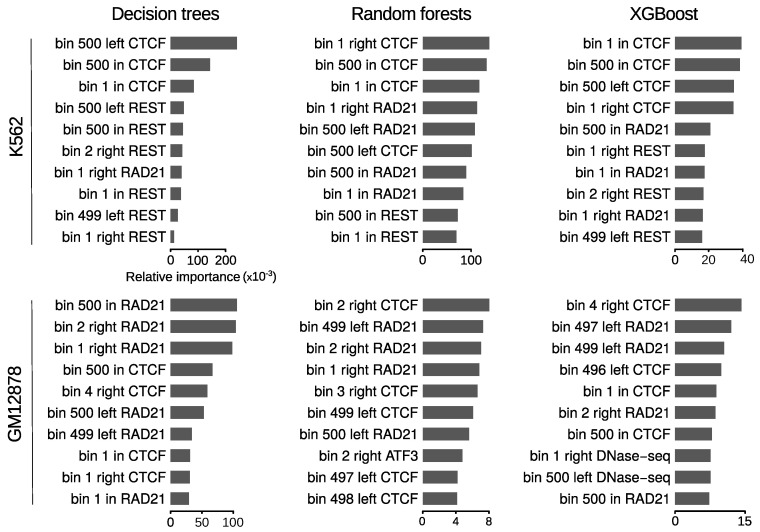
Ranking of top 10 important features for the prediction of RAD21 chromatin loops. Horizontal bars represent relative importances of featured bins. The terms ’left’, ’in’ and ’right’ are used for bins from 1 to 500, 501 to 1000 and 1001 to 1500, respectively. The relative position of the bins within one of these 3 windows is also included in the feature names.

**Figure 4 genes-11-00985-f004:**
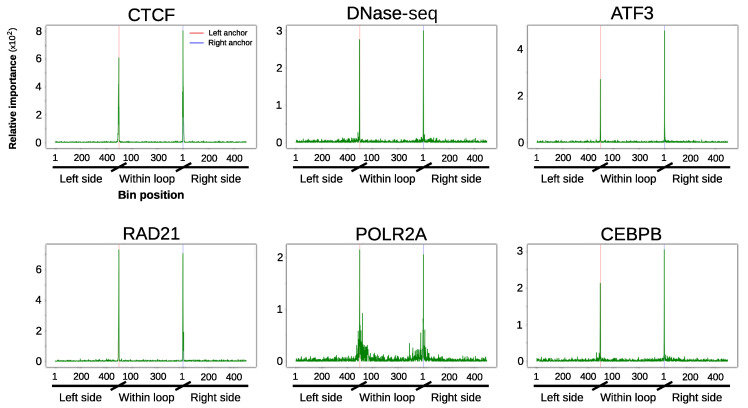
Position specific importance of selected high-throughput sequencing datasets in GM12878 cell line. Random forests importance score is shown for the 1500 bins corresponding to the most informative experiments. Coordinates of the x-axis are similar to those of Figure 5. Appendix A display similar plots for all the tested datasets as well as for Decision Trees and XGBoost algorithms. CTCF: CCCTC-binding factor.

**Figure 5 genes-11-00985-f005:**
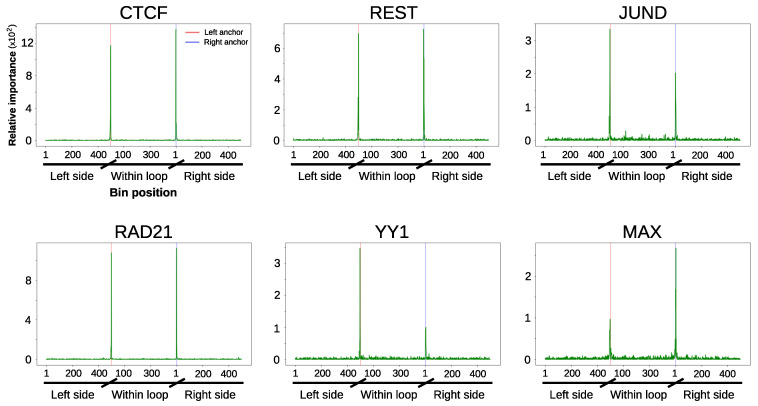
Position specific importance of selected high-throughput sequencing datasets in K562 cell line. Random forests importance score is shown for the 1500 bins corresponding to the most informative experiments. Coordinates of the x-axis are similar to those described in Figure 3, with left and right anchor position represented as red and blue vertical lines, respectively. Appendix A display similar plots for all the tested datasets as well as for Decision Trees and XGBoost algorithms.

**Figure 6 genes-11-00985-f006:**
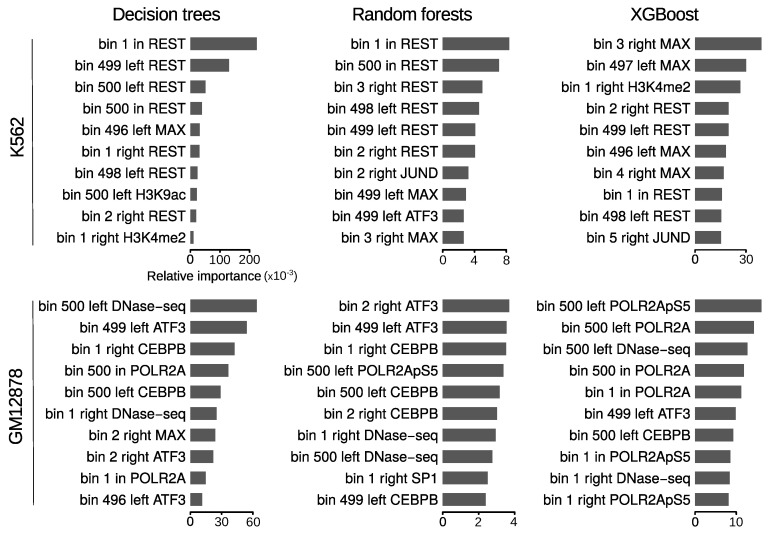
Ranking of top 10 important features for the prediction of RAD21 chromatin loops using only features associated with transcription. Horizontal bars represent relative importances as in Figure 3.

**Figure 7 genes-11-00985-f007:**
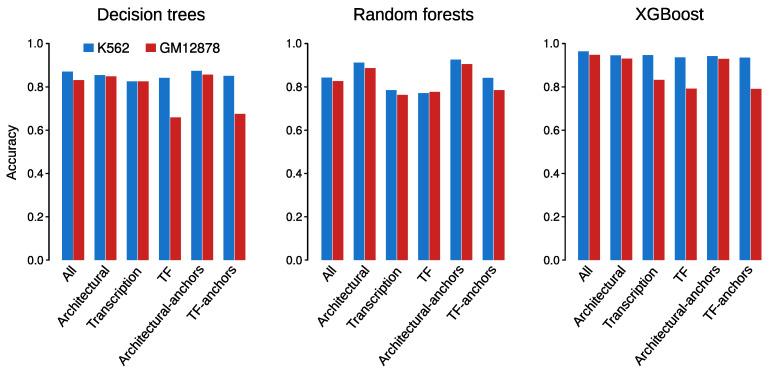
Accuracies of the Decision trees (DT), Random forests (RF) and XGBoost models trained with specific subset of chromatin features in K562 (blue) and GM12878 (red).

**Figure 8 genes-11-00985-f008:**
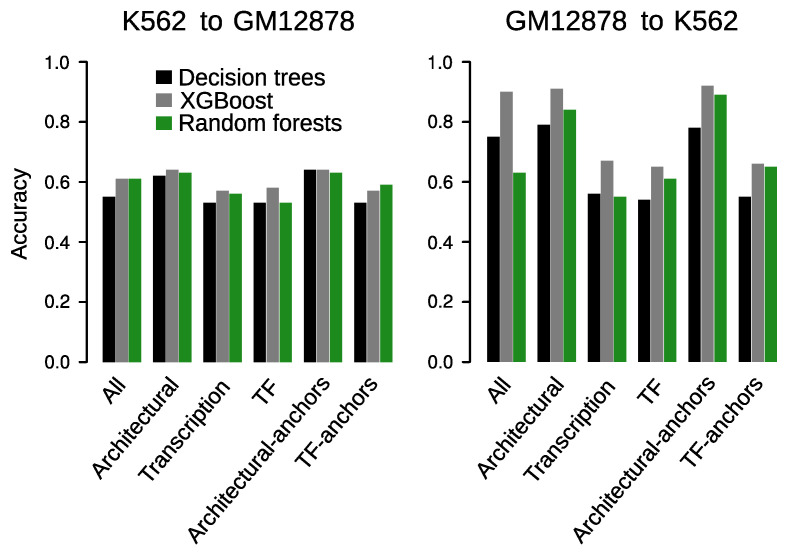
Cross cell lines accuracies of the DT, RF and XGBoost models trained with specific subset of chromatin features.

**Table 1 genes-11-00985-t001:** Machine learning performance of the proposed models in K562 cell line.

Algorithm	Accuracy	Precision	Recall	F1-Score
Decision Trees	0.8698	0.8707	0.8699	0.8698
Random Forests	0.8424	0.8469	0.8425	0.8419
XGBoost	0.9634	0.9638	0.9635	0.9635
SVM	0.8219	0.8224	0.8219	0.8218
MLP	0.8226	0.8231	0.8227	0.8226
Deep learning ANN	0.7930	0.7944	0.7930	0.7928

**Table 2 genes-11-00985-t002:** Machine learning performance of the proposed models in GM12878 cell line.

Algorithm	Accuracy	Precision	Recall	F1-Score
Decision Trees	0.8313	0.8314	0.8313	0.8313
Random Forests	0.8262	0.8284	0.8263	0.8261
XGBoost	0.9474	0.9485	0.9475	0.9474
SVM	0.8087	0.8088	0.8088	0.8087
MLP	0.8322	0.8328	0.8323	0.8322
Deep learning ANN	0.8064	0.8065	0.8065	0.8065

**Table 3 genes-11-00985-t003:** Machine learning performance of models trained without architectural factors information in K562 cell line.

Algorithm	Accuracy	Precision	Recall	F1-Score
Decision Trees	0.8257	0.8259	0.8257	0.8257
Random Forests	0.7846	0.7877	0.7846	0.7840
XGBoost	0.9467	0.9469	0.9467	0.9467
SVM	0.7264	0.7264	0.7237	0.7228
MLP	0.7168	0.7202	0.7169	0.7157
Deep learning ANN	0.6872	0.6873	0.6872	0.6871

**Table 4 genes-11-00985-t004:** Machine learning performance of models trained without architectural factors information in GM12878 cell line.

Algorithm	Accuracy	Precision	Recall	F1-Score
Decision Trees	0.6806	0.6817	0.6806	0.6805
Random Forests	0.7626	0.7648	0.7627	0.7624
XGBoost	0.8327	0.8334	0.8327	0.8327
SVM	0.7046	0.7068	0.7046	0.7042
MLP	0.7018	0.7061	0.7018	0.7008
Deep learning ANN	0.6608	0.6613	0.6608	0.6608

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
