# Peer review of "A Comparative Study of Supervised Machine Learning Algorithms for the Prediction of Long-Range Chromatin Interactions"

_genes, 2020, doi:10.3390/genes11090985_

Round 1

Reviewer 1 Report

1. The study was to “use publicly available 1D sequencing signals to model chromatin interactions”, or “for prediction of long-range chromatin interactions” as stated in the title. However, until section 2.2 for Materials and Methods, the authors mentioned “identification of RAD21-associated loops”. To avoid misleading readers, the machine learning problem as well as the function of RAD21 should be clearly described in the Abstract and/or Introduction. Why were only RAD21-associated loops used to model chromatin interactions?

2. As a routine procedure for model construction, the authors tested different machine learning algorithms, and found that XGboost might perform best. However, it is unclear whether the other models were well tuned. It is also possible that the performance of these learning algorithms might vary on different datasets or for slightly different problems. Thus, further work may be needed to establish the most suitable machine learning approach for predicting chromatin interactions. Especially, deep learning techniques have recently been shown to work well for predicting chromatin interactions.

Author Response

We would like to express our thanks for Reviewer #1’s feedback and helpful comments. We have revised the manuscript according to his/her (and other reviewers) suggestions, which we think have significantly improved its content. Please find below our responses:

1. The study was to “use publicly available 1D sequencing signals to model chromatin interactions”, or “for prediction of long-range chromatin interactions” as stated in the title. However, until section 2.2 for Materials and Methods, the authors mentioned “identification of RAD21-associated loops”. To avoid misleading readers, the machine learning problem as well as the function of RAD21 should be clearly described in the Abstract and/or Introduction. Why were only RAD21-associated loops used to model chromatin interactions?

We would like to clarify that the main purpose of our study is to perform comparisons of well-known machine learning approaches for the prediction of chromatin interactions. We decided to use RAD21 mediated loops because, to the best of our knowledge, and unlike CTCF loops, they had not been used before for this specific task. In any case, since RAD21 and CTCF highly co-localize at the borders of TADs as well as at loop anchors, ChIA-PET datasets targeting either protein do not differ much. We think that modeling loops associated to other proteins, or just loops derived from HiC data with no protein binding bias could be also a very interesting point to be further explored.

We followed Reviewer #1 suggestion and included specific information in the manuscript in which we highlight the importance of cohesin in chromatin wiring (lines 45-54). We also added text to clarify that we used RAD21 associated loops (lines 11,68), and included a ChIP-seq enrichment analysis showing RAD21 (and CTCF) signals at the anchors of selected loops (Figure S1).

2. As a routine procedure for model construction, the authors tested different machine learning algorithms, and found that XGboost might perform best. However, it is unclear whether the other models were well tuned. It is also possible that the performance of these learning algorithms might vary on different datasets or for slightly different problems. Thus, further work may be needed to establish the most suitable machine learning approach for predicting chromatin interactions. Especially, deep learning techniques have recently been shown to work well for predicting chromatin interactions.

We do agree with Reviewer #1 that this “top” of the various algorithms can vary depending on the type of task given to the algorithms and on what the data used to train the models looks like. This is why the experiment was run in parallel on two different cell lines, to avoid, at least a bit, this bias related to the data. However, for our given problem and datasets, the top ranking algorithm remains XGBoost. For the Decision trees and Random Forests, we have used Scikit-Learn, and we have tinkered with the metaparameters available for of each of these algorithms up to a point. We have done our best to avoid overfitting, and have decided to make a compromise between the training and testing scores in order to obtain the final metaparameters. Regarding XGBoost, it seems to be quite good, in our case, in handling the data out-of-the-box, and tinkering with its parameters a bit, such as eta, max_depth, gamma, subsample didn’t seem to have much effect when comparing with the default tuning; it could perhaps increase a bit the training score while reducing the testing score, so in the end we have decided to just leave it as it is.

In order to follow Reviewer #1 suggestion, we have included a deep learning artificial neural network to our analyses. Accordingly, we have extended the material and methods section to incorporate a description of this new method (lines 175-183) as well as how we have tuned the corresponding parameters (lines 184-201). We have also added the new results to Tables 1-4 and to the text of section 3.3. We have also modified Figure 2 accordingly.

Reviewer 2 Report

Vanhaeren et al. compared the performance of 5 popular machine learning algorithms in predicting long term chromatin interactions in two human cell lines. They found that gradient boosting outperforms the other four algorithms in predicting long-range interactions, and detected (as other studies reported before) that chromatin features close to the anchors cover most of the predictive information. Also, they highlight cell-type-specific binding of transcription factors at the anchors as essential determinants of chromatin wiring. To the best of my knowledge, there are dozens of in silico methods that have been proposed to predict chromatin interactions using diverse omics datasets, and this study is not different in that sense. The main added value of this study is the comparative analysis that was done to evaluate the performance of different algorithms. The manuscript is well written and deals with an important topic. I have only one major comment and one minor.

Major issue

As the authors emphasize in their manuscript, computational predictions based on publicly accessible sequencing datasets have emerged as an alternative strategy to generate virtual long-range chromatin interaction maps for new cell types for which experimental maps are not yet available. However, in their study, the authors compared the ability of the different machine learning algorithms to predict interaction in the same cell type of which the model was trained. It would be highly valuable to evaluate the accuracy of the different algorithms in predicting long term interactions of new cell types. For example, a model can be trained on one cell type and tested on a different cell type. Also, it would be essential to know which features are most important for each algorithm in predicting interactions in new cell types. Hence I advise the authors to add these analyses that have more practical implications than the analysis currently described in the manuscript.

Minor comment

The authors included in their model data of nine TFs, the logic behind choosing these specific TFs should be explained in the manuscript.

Author Response

Reviewer #2

We would like to express our thanks for Reviewer #2’s feedback and helpful comments. We have revised the manuscript according to his/her (and other reviewers) suggestions, which we think have significantly improved its content. Please find below our responses:

Major issue

As the authors emphasize in their manuscript, computational predictions based on publicly accessible sequencing datasets have emerged as an alternative strategy to generate virtual long-range chromatin interaction maps for new cell types for which experimental maps are not yet available. However, in their study, the authors compared the ability of the different machine learning algorithms to predict interaction in the same cell type of which the model was trained. It would be highly valuable to evaluate the accuracy of the different algorithms in predicting long term interactions of new cell types. For example, a model can be trained on one cell type and tested on a different cell type. Also, it would be essential to know which features are most important for each algorithm in predicting interactions in new cell types. Hence I advise the authors to add these analyses that have more practical implications than the analysis currently described in the manuscript.

We agree with Reviewer #2 that having information on cross cell line predictions would be quite valuable. Therefore, we have performed such analyses, which we have included in a new section called “3.7. Prediction across cell types”. As suggested by Reviewer #2, we have also studied the prediction abilities of the different subsets of features from section 3.6. The results of this analysis have been summarized in a new figure (Figure 8).

Minor comment

The authors included in their model data of nine TFs, the logic behind choosing these specific TFs should be explained in the manuscript.

The selection of these and not other TFs was restricted to those available in ENCODE that were common to both cell lines. We have included a comment in which we explain our motivation when selecting the sequencing datasets (lines 116-117).

Round 2

Reviewer 2 Report

The authors took into consideration all of my initial comments and improved their manuscript accordingly. They also added deep learning to the list of compared algorithms. I have just one comment: they described two algorithms 1. Multilayer perceptron which is a subset of deep-learning ANN (especially when using a high number of hidden layers as they did) and 2. Deep learning ANN. I’m not sure why they described them in separate sections. They should describe them under the same section of “Deep Learning”.

Author Response

we thank the reviewers for the useful comment. In order to address it we have indeed merged the two sections into a single one, 2.4.4. Deep Learning, where Multilayer perceptron and Deep Learning are described.